# Risk Factors for Elevated Serum Lipopolysaccharide in Acute Dengue and Association with Clinical Disease Severity

**DOI:** 10.3390/tropicalmed5040170

**Published:** 2020-11-16

**Authors:** N. L. Ajantha Shyamali, Sameera D. Mahapatuna, Laksiri Gomes, Ananda Wijewickrama, Graham S. Ogg, Gathsaurie Neelika Malavige

**Affiliations:** 1Centre for Dengue Research, University of Sri Jayewardenepura, Nugegoda 10250, Sri Lanka; ajaliyanage@yahoo.com (N.L.A.S.); smahapatuna@gmail.com (S.D.M.); laksiri79@gmail.com (L.G.); graham.ogg@ndm.ox.ac.uk (G.S.O.); 2National Institute of Infectious Diseases, Angoda 10620, Sri Lanka; anandawijewickrama012@gmail.com; 3MRC Human Immunology Unit, MRC Weatherall Institute of Molecular Medicine, NIHR Biomedical Research Centre, Oxford OX3 9DS, UK

**Keywords:** dengue, lipopolysaccharide, severe dengue, metabolic diseases, CRP

## Abstract

Although serum lipopolysaccharide (LPS) was shown to associate with development of severe dengue, the reasons for high LPS and its subsequent involvement in disease pathogenesis are not known. We assessed serum LPS, C-reactive protein (CRP), and procalcitonin in patients with acute dengue fever (DF = 129) and dengue haemorrhagic fever (DHF = 64) and correlated these observations with the presence of comorbid illnesses, and clinical disease severity. Serum LPS levels were significantly (*p* = 0.01) higher in patients with DHF, compared to those with DF. In total, 45 (70%) of those with DHF and 63 (49%) of those with DF had detectable LPS and therefore, the presence of LPS was significantly associated with DHF (*p* = 0.005, OR = 2.48, 95% CI: 1.29 to 4.64). Those with metabolic diseases, 22/29 (75.9%) and those with atopic diseases 17/22 (77.3%) were significantly more likely to have detectable LPS levels (*p* = 0.025, OR = 2.9, 95% CI-1.17 to 7.59 and *p* = 0.039, OR = 3.06, 95% CI-1.07 to 7.81 respectively). Those with detectable LPS levels were also more likely to develop shock and severe thrombocytopenia. Patients with detectable LPS were more likely to have elevated CRP levels and were more likely to develop DHF. Procalcitonin levels too were significantly (*p* = 0.009) higher in those with DHF compared to those with DF and were more likely to be high in those with detectable serum LPS. Since serum LPS levels were higher in patients with DHF and significantly more likely to be present in those with comorbid illnesses, the possible role of LPS in disease pathogenesis should be further investigated.

## 1. Introduction

Dengue viral infections represent one of the most important emerging mosquito-borne viral infections, resulting in approximately 100 million symptomatic infections annually [1]. Over 70% of these infections occur in Asia [1]. Although the mortality rates due to dengue have declined in South Asia, the incidence has markedly increased from 285.3 per 100,000 individuals in 1990 to 1371.1 in 2013 [2]. The estimated annual global cost of dengue is 8.9 billion $ [3], which is a huge burden to resource-poor developing countries. As there is no specific treatment for dengue, intense monitoring for complications and meticulous fluid management is currently the only option in the management of dengue. 

While the majority of individuals who are infected with the dengue virus (DENV) develop asymptomatic or an undifferentiated febrile illness, it can cause severe clinical disease manifestations, such as dengue haemorrhagic fever (DHF) and organ involvement, in 10% to 25% of individuals [4,5]. Weg de Van et al. showed that serum lipopolysaccharide (LPS) levels were significantly higher in those with plasma leakage and the LPS levels were associated with immune activation [6]. LPS is a potent stimulus for production of platelet-activating factor (PAF) and other inflammatory cytokines from monocytes [7,8]. In our earlier studies, we found that LPS appeared to act synergistically with the DENV by inducing significantly higher levels of PAF in DENV-infected monocytes than in uninfected monocytes [9]. Since PAF was found to be an important mediator of vascular leak [10], it is possible that LPS acts with the DENV in acute dengue, initiating or potentiating PAF-induced vascular leakage. This could further worsen intestinal barrier dysfunction, possibly leading to concurrent bacteraemia. 

Metabolic diseases, such as the presence of diabetes and hypertension, have shown to be risk factors that are associated with the development of severe dengue [11] and diabetes has been shown to be an independent risk factor for the development of shock in acute dengue [12]. We previously reported that the activity of secretory phospholipase A2 (sPLA2) was significantly higher in patients with DHF compared to those with DF [13]. LPS is one of the most potent mediators that potentiates sPLA2 [14]. Since sPLA2 activity was found to be highest during early illness, it is possible that pre-existing LPS in patients with comorbid illnesses, such as diabetes and obesity, possibly due to metabolic endotoxaemia, could lead to activation of sPLA2 and also contribute to generation of PAF. As patients with metabolic diseases have been shown to have low-grade endotoxaemia [15], presence of LPS at the time of infection with the DENV could aggravate disease severity. As the gut barrier permeability is increased in metabolic diseases, such as diabetes, gut bacteria could be an important source of LPS. Therefore, it would be important to determine if low-grade endotoxaemia seen in patients with metabolic diseases [16] contributes to severe dengue and subsequently concurrent bacteraemia. 

In this study, we sought to investigate if the risk factors associated with the presence of LPS in patients with acute dengue, and if the presence of LPS was associated with severe clinical disease. In addition, we also investigated the relationship between C-reactive protein (CRP) and procalcitonin, which are usually elevated in bacterial infections, as an indicator for the presence of LPS and severe clinical disease. 

## 2. Materials and Methods

### 2.1. Patients

We recruited 193 adult patients with confirmed acute dengue infection who were admitted to the National Institute of Infectious Diseases Sri Lanka during the years 2016 to 2018. Informed written consent was taken from all patients. Day one of illness was considered as the first day of fever. Clinical features, such as fever, abdominal pain, vomiting, bleeding manifestations, hepatomegaly, blood pressure, and urine output, were recorded several times a day in all patients. The full blood counts were monitored several times a day (three to four times) and liver function tests and ultrasound examination to detect the presence of fluid were carried out once a day. The clinical disease severity was classified according to the 2011 World Health Organization (WHO) guidelines [17]. Accordingly, patients who had ultrasound evidence of fluid leakage or a rise in the haematocrit of ≥20% from their baseline level were classified as having DHF. 

### 2.2. Ethics Approval

The study was conducted in accordance with the Declaration of Helsinki, and the protocol was approved by the Ethics Committee of University of Sri Jayewardenepura. Application number: 741/13. All individuals who participated gave informed written consent. 

### 2.3. Quantification of Serum LPS Levels

Serum was separated under sterile conditions and stored at −80°C until being tested for LPS. The LPS levels were determined with a commercially available ELISA assay according to the manufacturer’s instructions (Human LPS ELISA kit, CUSABIO, Wuhan, China). Serum samples were diluted to 1:2 with sample diluent and heat inactivated at 70°C for 20 min prior to use in the assay according to the manufacturer’s instructions. The assays were carried out twice in the initially recruited 80 patients and the values were found to be reproducible. 

### 2.4. Detection of Procalcitonin (PCT) and CRP 

PCT levels were determined in serum samples obtained daily from the time of admission to discharge, using a commercially available ELISA for PCT (Abcam, Cambrige, UK). The ELISAs were performed and results were interpreted according to the manufacturer’s instructions. The CRP levels were assessed by a quantitative immunoturbidimetric method using the Thermo Scientific^TM^ Indiko^TM^ system. CRP levels were expressed as mg/L and levels < 10 mg/L considered as normal. 

### 2.5. Serotyping of the DENV and Quantifying the Viral Loads

Serotyping of the DENV and quantifying of viral loads were carried out as previously described [4]. Extraction of viral RNA in serum was carried out using a QIAamp Viral RNA Mini Kit (Qiagen, Germantown, MD, USA) and transcribed to cDNA using a High Capacity cDNA reverse transcription kit (Applied Biosystems, Waltham, MA, USA) according to the manufacturer’s protocol. Quantitative real-time PCR was performed using the CDC real-time PCR assay for detection of the dengue virus and oligonucleotide primers and a dual-labelled probes for DEN 1−4 serotypes were used (Life Technologies, Carlsbad, CA, USA) based on published sequences [18]. 

### 2.6. Statistical Analysis

Statistical analysis was performed using GraphPad Prism version 7. As the data were not normally distributed, non-parametric statistical methods were used for data analysis. The differences in LPS, CRP, and PCT in single samples were done using the two tailed Mann–Whitney U-test. The degree of association between serum LPS levels and other markers was analysed using the Spearman correlation coefficient test. The degree of association between the presence of metabolic disease, the presence of LPS, and clinical disease severity was expressed as the odds ratio (OR), which was obtained from standard contingency table analysis by Haldane’s modification of Woolf’s method.

## 3. Results

### 3.1. Patient Characteristics

Of the 193 patients, 64 (33%) had DHF and 129 (67%) had DF based on the 2011 WHO dengue disease classification [17]. The mean age in those with DHF was 30.5 years (SD ±13.3) and 32.8 years (SD ± 14.9) in those with DF. The average day of recruitment of patients to the study was on day 4 (SD ± 1) of illness. The clinical and laboratory features of these 193 patients are shown in Table 1. There were no fatalities and only seven (3.6%) developed shock. 

### 3.2. Serum LPS Levels in Patients with Acute Dengue

We found that serum LPS levels were significantly (*p* = 0.01) higher in patients with DHF (median-11.36, IQR 0 to 28.5 pg/mL), when compared to those with DF (median-0, IQR 0 to 17.63 pg/mL). LPS was detected in 45 (70%) of those with DHF and 63 (49%) of those with DF (Figure 1). Therefore, patients with DHF were significantly more likely to have detectable LPS in their sera compared to those with DF (*p* = 0.009, OR = 2.3, 95% CI: 1.2 to 4.3) (Table 2). Among the 64 patients with DHF, there were 7 patients who developed shock (DSS). There was no significant difference (*p* = 0.62) between the LPS levels in patients with DHF who did not develop shock (median = 11.28, IQR- 0 to 29.23 pg/mL) compared to those who developed DSS (median-13.99, IQR 8.25 to 22.91 pg/mL) (Figure 1). However, although not significant, those with LPS were more likely to develop shock (Table 2). 

We also proceeded to determine if higher viral loads associate with clinical disease severity and the relationship with the viral loads with serum LPS levels. The viral loads in those with DF (median 302, IQR 0 to 11,430 viral copies/mL) were similar to those with DHF (median 324, IQR 0 to 2190 viral copies/ml, *p* = 0.48). Interestingly, we observed a statistically significant but weak correlation between the degree of viraemia and serum LPS levels (Spearmans’ r = −0.28, *p* = 0.004).

### 3.3. Association of LPS Levels with Presence of Comorbid Illnesses

As patients with metabolic diseases have been shown to have low-grade endotoxaemia [15], presence of LPS at the time of infection with the DENV could aggravate disease severity. Therefore, in order to find out if patients with metabolic diseases had higher LPS and more severe disease, we analysed the association of serum LPS levels with the presence of comorbid illnesses. In total, 47/193 (24%) patients in our cohort had co-morbid illnesses. In total, 22/47 (47%) had atopic diseases, such as asthma and allergic rhinitis, and 29/47 (62%) of them had metabolic diseases, such as hypertension, diabetes, ischaemic heart disease, and hyperlipidaemia. In total, 22/29 (75.9%) patients with metabolic diseases and 17/22 (77.3%) of patients with atopic diseases had detectable LPS in their sera. Of those with metabolic diseases (*n* = 29) and who had LPS, 2/29 had DHF and 20/29 had DF. Of the 22 atopic patients with dateable LPS in serum, 3/22 had DHF and 14/22 had DF. Therefore, 34/63 (53.9%) patients who had detectable LPS in DF had comorbid illnesses. Therefore, those with metabolic diseases were significantly more likely (*p* = 0.024) to have detectable LPS than those who did not (OR = 2.9, 95% CI- 1.17 to 7.59) and those with atopic diseases (asthma and allergic rhinitis) were also significantly more likely (*p* = 0.04) to have detectable LPS in their sera compared to those without atopic disorders (OR = 3.1, 95% CI-1.1 to 7.8) (Table 2). However, those with metabolic disease or atopic disease were not more likely to develop DHF in our cohort, but this study was not powered to evaluate such associations. 

### 3.4. Association between Serum CRP and LPS in Patients with Acute Dengue

Since patients with DHF were more likely to have higher LPS levels, we next sought to investigate the possible association of CRP values with serum LPS. We found that although CRP levels were slightly higher in patients with DHF (median 8.95, IQR 6.5 to 18.65 mg/L) than those with DF (median 7.1, IQR 5.5 to 15.3 mg/L), the difference was not significant (*p* = 0.52) (Figure 2). In addition, there was no significant difference (*p* = 0.29) between the CRP levels in patients with DHF who did not develop shock (median-8.95, IQR 6.5 to 18.65 mg/L) compared to those who developed DSS (median-15.05, IQR 6.49 to 38.15 mg/L). However, patients with elevated CRP levels (>10 mg/L) were significantly more likely (*p* = 0.01) to have detectable LPS in their sera compared to those with normal CRP levels (OR = 2.261, 95% CI: 1.206 to 4.169). There was no significant correlation between CRP levels and viral loads (Spearman’s r = 0.06, *p* = 0.56).

A high CRP in early illness was suggested to have a good predictive value in developing severe dengue [19,20]. Although CRP levels were not significantly high in patients with DHF compared to patients with DF, in those who were recruited ≤4 days of illness (before development of vascular leakage), 9/24 (37.5%) who proceeded to develop DHF and 5/40 (12.5%) of those who developed DF had CRP values > 20 mg/L). Therefore, CRP values of >20 mg/L in early illness was associated with a significantly higher risk (*p* = 0.02) of subsequently developing DHF (OR 4.2, 95% CI 1.28 to 13.44). 

### 3.5. Serum Procalcitonin (PCT) Levels in Patients with Acute Dengue

PCT is considered to be a biomarker that can be useful in differentiating sepsis from other non-infection triggers in critically ill patients [21]. However, it was recently shown that PCT can also be elevated in patients with acute dengue and that PCT levels more than 0.7 ng/mL were associated with DSS [22]. PCT levels above 0.1 ng/mL have been shown to indicate probable sepsis and therefore patients with PCT values over 0.1 ng/mL are considered to have an elevated PCT [23,24]. Therefore, we proceeded to find out the usefulness of PCT in in identifying those who develop complications of acute dengue (DHF, DSS) and its association with LPS. 

Serum PCT levels were significantly (*p* = 0.009) higher in those with DHF (median-0.13 ng/mL, IQR-0.07 to 0.20 ng/mL) compared to those with DF (median-0.08 ng/mL, IQR-0.04 to 0.13 ng/mL) (Figure 3). PCT were elevated (>0.1 ng/mL) in 28 (44.4%) patients with DHF and 38 (29.2%) patients with DF, and although not significant (0.05), it was associated with the presence of DHF (OR 1.9, 95% CI: 1.0 to 3.6). There was no significant difference (*p* = 0.64) between the PCT levels in patients with DHF who did not develop shock compared to those who developed DSS (Figure 3). In addition, none of the patients in our cohort with DSS had PCT values > 0.7 ng/mL. Although not significant (*p* = 0.09), those who had detectable PCT levels were more likely to have detectable LPS than those who did not (OR = 1.7, 95% CI 0.94 to 3.2). Serum PCT levels also statistically significantly but weakly correlated with LPS levels (Spearman’s r = 0.163, *p* = 0.023) and CRP levels (Spearman’s r = 0.233, *p* = 0.001). 

We also found that 20/29 (69%) of those with metabolic diseases were significantly more likely (*p* = 0.004) to have elevated PCT levels in their sera (OR = 3.5, 95% CI- 1.47 to 8.5). However, there was no association between elevated PCT and the presence of atopic disorders (asthma and allergic rhinitis). 

In total, 31/193 (16%) patients with acute dengue had received antibiotics due to clinical suspicion of a concurrent bacterial infection (recurrence of fever, productive cough, and sore throat), during the later stages of illness. Of these patients, 10/31(32%) had positive bacterial cultures and the predominant organisms identified were *Escherichia coli*, *Streptococcus*, *Staphylococcus* species, and *Pseudomonas aeruginosa*. In total, 15/31 (48%) patients with suspected concurrent bacteraemia and 7/10 (70%) patients with culture-confirmed bacteraemia had DHF. Of those with DSS, 2/7 (28%) had suspected concurrent bacteraemia and only 1/7 had culture-confirmed bacteraemia. Only 5/10 (50%) patients with culture-confirmed bacteraemia had detectable levels of LPS and elevated levels of PCT in their serum.

## 4. Discussion

In this study, we found that serum LPS levels were significantly higher in patients with plasma leakage (DHF) as previously reported in other cohorts [6]. Although LPS was detected at a higher frequency of in patients with DHF (70%), it was also detected in 49% of patients with DF, who did not have any features of plasma leakage. Those who were more likely to have endotoxemia were those with either metabolic diseases (75.9%) or those with atopic diseases, such as allergic rhinitis or asthma (77.3%). In fact, 53.9% of patients with DF who had detectable LPS had metabolic disease or atopy. As patients with metabolic diseases have been shown to have low-grade endotoxaemia [15,25], it is possible that LPS detected in these patients were due to the presence of pre-existing LPS, due to increased permeability of the gut barrier in patients with metabolic disease. As patients with diabetes and hypertension are known to be at a higher risk of developing severe dengue [12,26] and also a higher risk of fatalities [27], it is possible that LPS levels in such patients contribute to disease pathogenesis when infected with the DENV by increased immune activation. 

PAF is an inflammatory phospholipid, which was shown to be an important mediator of vascular leak, and PAF receptor blockade showed a reduction in the extent of leakage in acute dengue [10,28]. Since the DENV was shown to act synergistically with LPS to produce PAF from monocytes [9], it is possible that patients with pre-existing endotoxaemia are more likely to develop severe disease due to these mechanisms. Indeed, in this cohort, those with detectable LPS levels in serum were more likely to have complications, such as DHF, shock, and severe thrombocytopenia (platelet counts < 20,000 cells/mm^3^). High LPS levels have also been associated with many other viral infections. Certain variants of the Polio virus have been shown to bind to LPS, enhancing environmental stability and attachment to host cells, which contributes to viral virulence [29]. On the other hand, LPS binding to the influenza virus changed its morphology, leading to destabilization of the virus [30]. Presence of LPS in HIV infection has shown to have a detrimental effect by causing immune activation, lowering response to therapy, increasing virus-unrelated comorbidities, and overall increasing the disease progression [31]. In our study, we observed that presence of LPS was associated with markers of immune activation, such as high CRP and PCT levels. However, the mechanisms of LPS interaction with the DENV should be studied to further understand the role of LPS in disease pathogenesis. 

CRP levels exceeding >30 mg/L during the first 3 days of illness were shown to be a possible biomarker in predicting the development of severe dengue [20]. In addition, CRP was also reported as a predictor of shock in acute dengue and it was shown that those with dengue shock syndrome had median CRP values of 124.5 mg/L, which are values usually only seen in patients with acute bacterial infections [19]. In our cohort, we too observed that high CRP levels during early illness appeared to be significantly associated with subsequent development of DHF. In addition, high CRP levels were associated with the presence of LPS in serum. Although previously it was reported that those with dengue shock syndrome had median CRP values of 124.5 mg/L [19], only one patient in our cohort had CRP values exceeding 50 mg/L. This patient, who had CRP levels between 120 and 150 during the critical period, did not develop shock, although he did develop acute liver failure. Therefore, although high CRP levels during early illness has been shown to associate with subsequent development of DHF, it was not a strong predictive marker of development of DHF in this cohort. 

PCT levels were significantly higher in patients with DHF compared to DF although the levels were not higher in those with DSS as previously reported [22]. However, PCT levels > 0.1 ng/mL were associated with the presence of DHF. In addition, the presence of PCT was shown to associate with the presence of LPS, and the levels of PCT correlated with LPS levels, suggesting that endotoxins could be leading to a rise in serum PCT. PCT has been shown to neutralize the activity of LPS, thereby reducing subsequent LPS-mediated immune activation [32]. LPS has been shown to be an important stimulus of PCT production, through Nuclear Factor-κb (NF-κb) activation [33]. Therefore, the increase in PCT in DHF and its association of LPS in dengue could be due to direct stimulation of production of PCT by LPS. 

## 5. Conclusions

We found that LPS levels were significantly higher in those with DHF compared to those with DF, and that 70% of those with DHF had detectable LPS, compared to 49% of those with DF as previously reported. Those with LPS were significantly more likely to have either metabolic disease (75.9%) or atopic diseases (77.3%). Those with detectable LPS levels were also more likely to develop shock and severe thrombocytopenia. Since patients with such comorbid diseases are at increased risk of developing severe and fatal dengue, the role of LPS in disease pathogenesis should be further investigated. 

## Figures and Tables

**Figure 1 tropicalmed-05-00170-f001:**
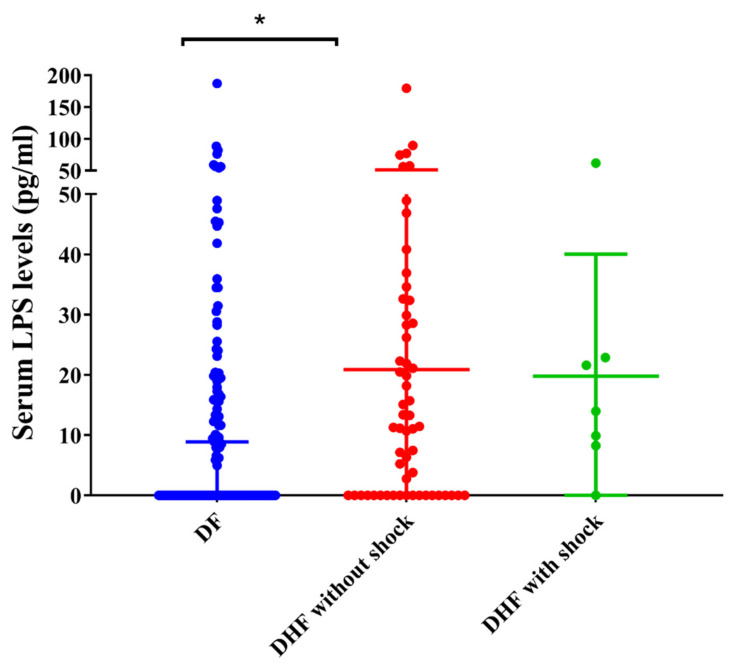
Lipopolysaccharide (LPS) levels in patients with acute dengue. LPS levels were measured by ELISA in patients with acute (A) DF (*n* = 129, indicated in red), DHF (*n* = 57, indicated in blue), and DSS (*n* = 7, indicated in green) during day 4 (SD ± 1) of illness. Error bars indicate the median and interquartile range (IQR). * *p* < 0.05.

**Figure 2 tropicalmed-05-00170-f002:**
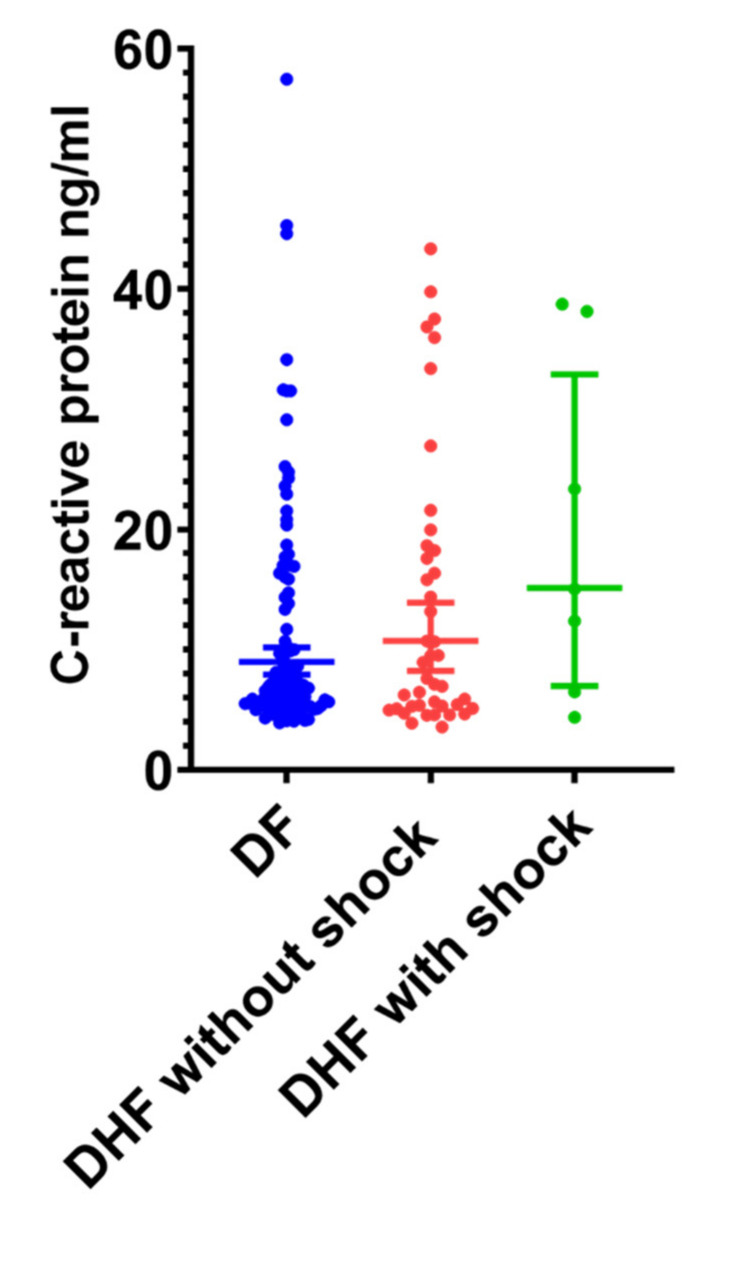
C-reactive protein (CRP) levels in patients with acute dengue. CRP levels were measured using the quantitative immunoturbidimetric method in patients with (A) DF (*n* = 129, indicated in blue) and DHF (*n* = 64, indicated in red) between day 4 and 6 of illness. Error bars indicate the median and interquartile range (IQR).

**Figure 3 tropicalmed-05-00170-f003:**
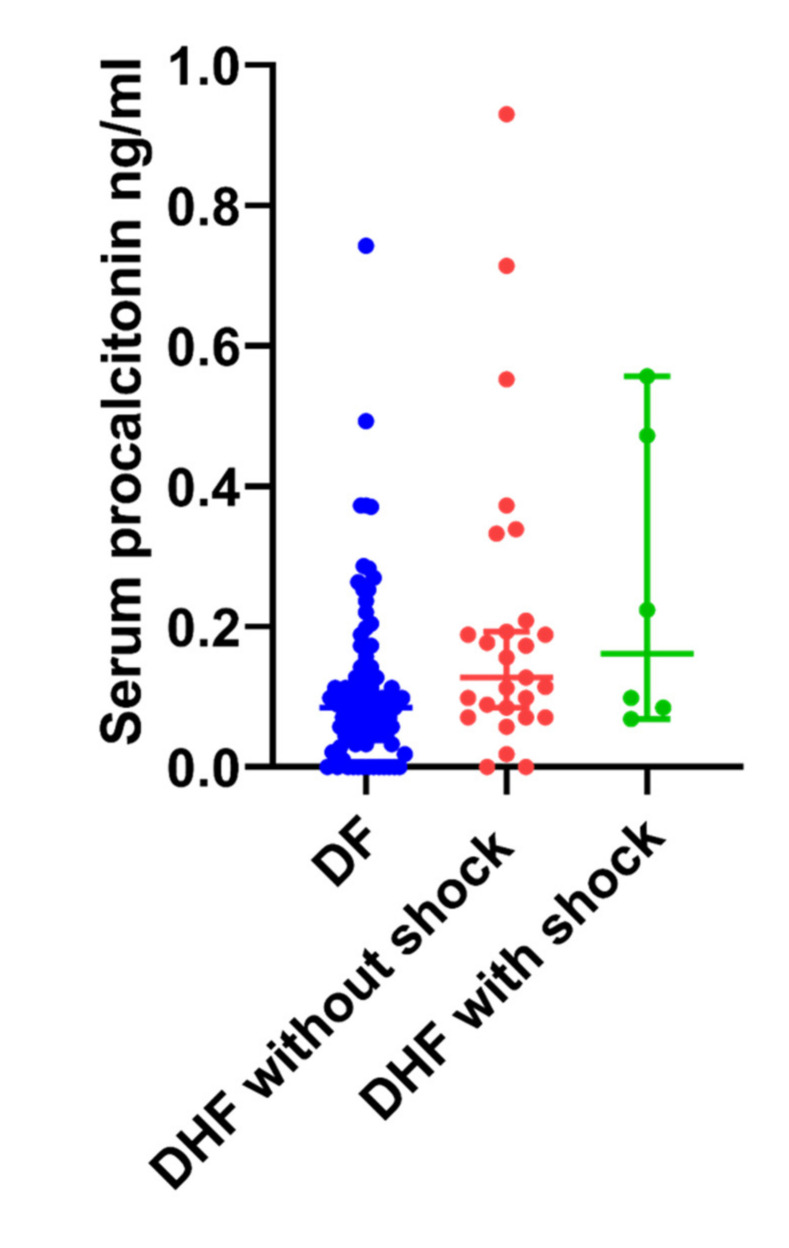
Serum procalcitonin (PCT) levels in patients with acute dengue. Serum PCT levels were measured by ELISA in patients with (A) DF (*n* = 129, indicated in blue) and DHF (*n* = 64, indicated in red).

**Table 1 tropicalmed-05-00170-t001:** Clinical and laboratory characteristics of patients with Dengue Haemorrhagic Fever (DHF) and Dengue Fever (DF) Recruited for the study.

	DHF (*n* = 64)	DF (*n* = 129)
**Clinical features**		
Vomiting (%)	15 (23)	21 (16)
Abdominal pain (%)	31 (48)	21 (16)
Diarrhoea (%)	13 (20)	27 (21)
Hepatomegaly (%)	18 (28)	1 (0.7)
Bleeding manifestations (%)	3 (5)	0
Pleural effusion (%)	26 (41)	0
Ascites (%)	56 (87)	0
Shock (%)	7 (11)	0
**WBC count**		
<4 × 10^9^/L (%)	41 (64)	94 (73)
**Lowest platelet count**		
<20,000 (%)	36 (56)	5 (4)
20,000–50,000 (%)	21 (33)	26 (20)
50,000–100,000 (%)	7 (11)	73 (56)
>100,000 (%)	0	25 (19)

**Table 2 tropicalmed-05-00170-t002:** Clinical and laboratory characteristics associated with presence of endotoxins.

Clinical Characteristic	Patients with Detectable LPS *n* = 107	Patients without LPS *n* = 86	Odds ratio (95% Confidence Intervals	*p* Value
DHF (*n* = 64)	44 (68.8%)	20 (31.2%)	2.3 (1.2 to 4.2)	0.009
Shock (*n* = 7)	6 (85.7%)	1 (14.3%)	7.2 (1.1 to 83.3)	0.05
Severe thrombocytopenia (<20,000 cells/mm^3^)	16 (80%)	4 (20%)	5.4 (1.8 to 15.3)	0.002
Metabolic disease (*n* = 29)	22 (75.8%)	7 (24.1%)	2.9 (1.2 to 7.6)	0.02
Allergic diseases (*n* = 22)	17 (77.3%)	5 (22.8%)	3.1 (1.1 to 7.8)	0.04
Any comorbidity (*n* = 47)	34 (72.3%)	13 (27.7)	2.6 (1.2 to 5.1)	0.01

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
