# Peer review of "Risk Factors for Elevated Serum Lipopolysaccharide in Acute Dengue and Association with Clinical Disease Severity"

_tropicalmed, 2020, doi:10.3390/tropicalmed5040170_

Round 1

Reviewer 1 Report

The manuscript is well written, with concise abstract, clear objectives with sufficiently described methodologies, and appropriate conclusions and references. Although it does not bring a large amount of new information, the work seems sound and deserves to be published. However, as previous works have detected an increase in LPS in the plasma of patients with DENV and correlated with the severity of the disease, we suggest that the authors include in the conclusions a sentence contemplating these studies as e.g. “our results confirm previous works of the presence of a high level of LPS in the plasma of dengue patients ”

Author Response

Comments: The manuscript is well written, with concise abstract, clear objectives with sufficiently described methodologies, and appropriate conclusions and references. Although it does not bring a large amount of new information, the work seems sound and deserves to be published. However, as previous works have detected an increase in LPS in the plasma of patients with DENV and correlated with the severity of the disease, we suggest that the authors include in the conclusions a sentence contemplating these studies as e.g. “our results confirm previous works of the presence of a high level of LPS in the plasma of dengue patients ”

Response: We thank the reviewer for carefully going through the manuscript and for the useful suggestions. As the reviewer has pointed out, previous reports have shown that high levels of LPS is detected in those with DHF. We have included a sentence under conclusions as well as suggested by the reviewer.

Reviewer 2 Report

Authors presented results are dealing with important question if the presence of LPS was associated with severe clinical diseases  - acute dengue fever (DF) and dengue haemorrhagic fever (DHF)=64).

From 193 enrolled patients 129 presented DF and 64 DHF.  Studied group is very heterologous - 47/193 (24%) 149 patients in our cohort had co-morbid illnesses –as Authors described :  “atopic diseases such as asthma, 150 allergic rhinitis and 29/47 (62%) of them had metabolic diseases such as hypertension, diabetes, 151 ischaemic heart disease and hyperlipidemia. 22/29 (75.9%). No control group for co-morbid illnesses are provided.

 Interesting  observation that presence of higher  LPS serum level was significantly associated with DHF need to be compared with similar group of patients with metabolic diseases but not infected by denga viral infection. Similar comparison need to be done for CRP and PCT.

 Sample serum collection were done only once? . Since LPS level as well as CRP and PCT amount might be time depended repetition of sample collection will allowed to have more solid data.  To make picture more complex  - 31/193 (16%) patients with acute dengue had received antibiotics and some of them had confirmed bacteriemia. Some antibiotic may increase endotoxin level in serum.

Major observation is : “We found that serum LPS levels were significantly (p=0.01) higher in patients with DHF 124 (median- 11.36, IQR 0 to 28.5 pg/ml), when compared to those with DF (median-0, IQR 0 to 17.63 125 pg/ml). LPS was detected in 45 (70%) of those with DHF and 63 (49%) of those with DF (Fig 1)”.

Although 28.5 versus 17,3 pg/ml for DHF and DF symptoms, respectively  are statistically significant  no information about reproducibility and  entotoxin test  sensitivity  are provided.

Author Response

Reviewer 2:

Comment: Authors presented results are dealing with important question if the presence of LPS was associated with severe clinical diseases - acute dengue fever (DF) and dengue haemorrhagic fever (DHF)=64). From 193 enrolled patients 129 presented DF and 64 DHF.  Studied group is very heterologous - 47/193 (24%) 149 patients in our cohort had co-morbid illnesses –as Authors described :  “atopic diseases such as asthma, 150 allergic rhinitis and 29/47 (62%) of them had metabolic diseases such as hypertension, diabetes, 151 ischaemic heart disease and hyperlipidemia. 22/29 (75.9%). No control group for co-morbid illnesses are provided.

Response: We apologize for lack of clarity. In the analysis of metabolic syndrome, atopic diseases, comorbidities, and their association with the presence of LPS, we carried out two analysis. We analyzed the presence of comorbidities in those who had LPS and who did not have detectable LPS. In table 2, we have described the proportion of individuals with LPS who had comorbidities and those who did not have detectable LPS and comorbidities (control group). Also, we have also analyzed the presence of comorbidities in those with DHF and DF. We have also described this between line 192 to 201. We have compared the proportion of those with metabolic disease having LPS vs the proportion of those without metabolic disease who had LPS (control group).

Comment 2: Interesting observation that presence of higher LPS serum level was significantly associated with DHF need to be compared with similar group of patients with metabolic diseases but not infected by dengue viral infection. Similar comparison need to be done for CRP and PCT.

Response: Thank you for this suggestion. It has been shown in many studies that those with metabolic syndrome, do have LPS in serum, due to gut microbial dysbiosis. The mechanisms by which pre-existing LPS in those with metabolic disease can give rise to DHF has been described in a recent review by us (Malavige et al, Fronteriers in Cellular Infection and Microbiology, 2020. https://www.frontiersin.org/articles/10.3389/fcimb.2020.590004/full )

The following references are included in the manuscript:

  1. Neves AL, Coelho J, Couto L, Leite-Moreira A, Roncon-Albuquerque R. Metabolic endotoxemia: a molecular link between obesity and cardiovascular risk. Journal of molecular endocrinology. 2013;51(2):R51-R64.
  2. Moludi J, Alizadeh M, Lotfi Yagin N, et al. New insights on atherosclerosis: A cross-talk between endocannabinoid systems with gut microbiota. J Cardiovasc Thorac Res. 2018;10(3):129-137.

Comment: Sample serum collection were done only once? . Since LPS level as well as CRP and PCT amount might be time depended repetition of sample collection will allowed to have more solid data.  To make picture more complex - 31/193 (16%) patients with acute dengue had received antibiotics and some of them had confirmed bacteriemia. Some antibiotic may increase endotoxin level in serum.

Response: Thank you for this important question. The samples were collected during early illness, when none of the patients were administered any antibiotics. The antibiotics were only given at a later stage of illness, when a secondary bacterial infection was suspected. We have included this information in the revised version of the manuscript.

Although collection of multiple samples throughout the course of illness, would have provided more information of the changes in LPS, CRP and PCR, it was beyond the scope of this study. However, all patients were followed up daily from admission until discharge from hospital.

Comments: Major observation is : “We found that serum LPS levels were significantly (p=0.01) higher in patients with DHF 124 (median- 11.36, IQR 0 to 28.5 pg/ml), when compared to those with DF (median-0, IQR 0 to 17.63 125 pg/ml). LPS was detected in 45 (70%) of those with DHF and 63 (49%) of those with DF (Fig 1)”.Although 28.5 versus 17,3 pg/ml for DHF and DF symptoms, respectively  are statistically significant  no information about reproducibility and  entotoxin test  sensitivity  are provided.

Response: Thank you for this important suggestion. We did repeat the LPS assay twice in approximately 80 patients and the results were reproducible. We have included this information under the results section in the revised version of the manuscript.

Round 2

Reviewer 2 Report

Authors information : "We did repeat the LPS assay twice in approximately 80 patients and the results were reproducible. We have included this information under the results section in the revised version of the manuscript." , 

Please provide data of asseys.